# Flexural and Shear Deformation of Basement-Clamped Reinforced Concrete Shear Walls

**DOI:** 10.3390/ma17102267

**Published:** 2024-05-11

**Authors:** Harald Schuler

**Affiliations:** School of Architecture, Civil Engineering and Geomatics, FHNW University of Applied Sciences and Arts Northwestern Switzerland, Hofackerstrasse 30, 4132 Muttenz, Switzerland; harald.schuler@fhnw.ch

**Keywords:** earthquake analysis, RC shear wall, load–displacement behaviour, tension shift, sliding shear, basement, clamping

## Abstract

For a precise analysis of buildings under earthquake effects, the load–deformation behaviour of the bracing walls must be comprehensively known. The horizontal bracing walls are often clamped in the basement; however, less attention has been paid to these walls’ clamping parts in the past. This study presents three shear wall experiments (NW 1, NW 2, NW 3) with heights up to six meters in a scale of 1:1.5 to the real size. Measured were the force–displacement curve, the curvature distribution over the height, the crack pattern, and the crack opening and spacing. Twelve displacement transducers, an optical measurement system with eight cameras, and a manual crack measurement were utilised. Out of the measurements, the impact of the tension shift effect on the load–displacement curves could be quantified for the cantilever part of the walls. Additionally, it was found out that a sliding failure in the clamping part must be considered if the aspect ratio of H/L is equal to or less than one.

## 1. Introduction

RC shear walls are one of the most effective structural elements for stabilising structures against horizontal earthquake actions. Several shear wall experiments have been carried out and numerous different effects studied in the last 30 years: Load deformation behaviour/curvature distribution/crack pattern [1,2,3,4,5,6,7,8,9,10]; strain penetration/anchorage slip/lap splicing [11,12]; tension shift [13]; and squat walls/sliding shear [14,15,16,17,18,19], with only a few mentioned here. Most experiments on RC shear walls are on cantilever walls; however, walls in building construction are often clamped into the basement and then also consist of a clamping wall section. Horizontal forces can thus be safely transferred to the foundation or subsoil.

In a simplified design, often only the cantilevered wall part of a shear wall is considered. The clamping part is often omitted to simplify the calculation. However, it provides additional deformation capacity but also can fail prematurely. Thus, it should be considered in a more precise analysis. Sliding shear failures were observed directly below the basement ceiling, where high bending moments and shear forces occur simultaneously. The shorter the clamping length, the greater the shear force. e.g., earthquakes in Chile caused a sliding shear failure under basement ceilings [17]. In Figure 1, it is sketched how a stiffening shear wall is clamped into the basement. The cutting forces, curvatures, and the deformations are illustrated, especially when sliding shear occurs under the basement ceiling. The special feature of the presented test configuration is that two sections, which are in a different kind of way stressed and therefore also deformed, can be analysed in interaction with each other. Both contribute to the total deformations. The maximum moments occur in the construction joints. Below the basement ceiling, it is slightly lower than above the basement ceiling because the moment decreases there more quickly. The section in the basement corresponds to a squat wall. The section of the upper floors is rather slender. There are studies in the literature for both wall types, squat and slender walls. However, there is a lack of information on the interaction of both wall sections with each other, resp. on the total behaviour of both sections together. Questions are as follows: How are the deformations distributed over both sections? Which kind of failure mode occurs? Where does the failure occur, in the cantilever or in the clamping part? There are no experimental studies on these questions in the literature available.

In three large-size experiments, load–deformation behaviour of RC concrete walls clamped into the basement was investigated, where the geometry and reinforcement ratios were varied. The focus was on the overall behaviour (cantilevered + clamped part), analysed on simplified specimens: simplified ceiling cutout, continuous longitudinal reinforcement, etc. 

## 2. Experimental Set-Up and Test Specimens

### 2.1. Experimental Set-Up

A test facility was developed to investigate shear walls clamped into the basement. One of the central goals of the test campaign was to study the influence of the clamped part on shear walls’ load–deformation behaviour. Figure 2 shows the test set-up, where clamping was realised between the middle and the bottom axis and force was applied on the upper axis. A challenge was to avoid additional clamping effects from vertical constraints and realise free rotation in the middle and bottom axes. With prestressed rods on all levels, horizontal supports were achieved in both directions, north and south. The wall’s vertical support was on the middle axis on the right side (north), where the vertical force was directly introduced in the adjacent steel hinge. The other five horizontal supports on the left and right side of the wall were freely movable vertically. The supports and the force are illustrated in red.

Figure 3 shows the front side of the shear wall in the test facility (top left) and the rear side with the measuring frame screwed onto the ceiling cutout (top right). The front side was used for optical measurement, the rear side for measurement with the displacement transducers. Figure 3 (bottom left) shows the measurement of the crack widths on the side of the walls; Figure 3 (bottom right) shows the sensors fixed on the frame and specimen. The measurement with the displacement transducers is a relative measurement system, with respect to the ceiling cutout. The optical measurement is an absolute measurement system because the cameras are fixed on a stationary wall.

### 2.2. Shear Walls

The investigation comprised three walls with varied reinforcement and aspect ratios. NW 1 (Figure 4) was designed to represent existing building walls, containing a small amount of reinforcement and no boundary reinforcing. The shear (horizontal) reinforcement was designed to ensure that no shear failure could occur, because the investigation focus was on flexural and sliding deformations. The thicker middle part (500 mm) represents the ceiling above the basement; the vertical reinforcement was continuous without lap splicing above the basement ceiling. An overview of the geometry and reinforcement ratios of all walls is given in Table 1. NW 2 (Figure 5) was designed to represent a ductile earthquake wall in a new building and contained boundary reinforcement over 0.15 times the wall length. The shear (horizontal) reinforcement was designed that way so that no shear failure could occur. Additional stirrups were inserted in the boundary reinforcement area; geometry was the same as for NW 1. NW 3 (Figure 6), like NW 1, also represented an existing wall of the same height but different length: 1.33 times longer than NW 1. The reinforcement ratio was practically identical to that of NW 1, except for horizontal reinforcement in the basement, which had to be sufficient to transfer greater shear forces.

In Table 1, the geometries and reinforcement ratios are summarised for all walls. Table 2 lists the concrete’s material properties, Table 3 the steel’s material properties. The walls were fabricated in four stages along the vertical axis. The concrete properties are given from stage 1 and 3, which corresponded to the concrete in the basement wall and cantilever wall part adjacent to the ceiling cutout. Mean value is given from at least five material tests; concrete compression strength was measured on cubes with a 150 mm edge length. As a basis for concrete tensile strength, splitting tensile strength was measured on cylinders with a 300 mm length and a 150 mm diameter. Reinforcement bar material properties were obtained from the bilinear approximation of the stress–strain relationship measured in the tension test.

### 2.3. Measurement Equipment

#### 2.3.1. Measurement with Inductive Displacement Transducers (Back Side)

Totally, 12 displacement transducers were used (Figure 7 (left)): A frame with six sensors was mounted above the ceiling cutout and a frame with—again—six sensors was mounted below the ceiling cutout. This means that displacements in the upper and lower floor are measured independently, each in relation to the ceiling cutout. Thus, the displacements of the cantilevered and clamping part are available separately, and their proportion of the total displacement can be specified. Figure 7 (left) shows the position of the measurement points and the measurement direction of the displacement transducers. With the horizontal displacement transducer at the top and on the bottom, the displacements of cantilevered and clamped part of the walls are measured. With the inner transducers, the sliding displacements in the construction joints are measured. The vertical displacements are not used in this publication. For example, the inner ones are used to measure the crack opening in the construction joint when analysing the sliding shear resistance. 

#### 2.3.2. Optical Measurement (Front Side)

Eight cameras were used to measure the displacement field over the total wall. Four of the cameras were more focused on the upper wall part and four on the lower wall part. The cameras were positioned between four to seven metres away from the wall. The system was able to measure displacements in the X-, Y-, Z-direction, where only the X-, Y-direction (perpendicular plane to the cameras) was needed. The reflector grid had a horizontal and vertical distance between the reflectors of 200 mm; Figure 7 (right) shows the grid. Reflector points on the ceiling cutout could not be used because they were partially obscured by the steel structure; thus, the distance between the upper and lower reflector points adjacent to the ceiling cutout was 0.8 m. Over this length, curvatures, strains, and displacements had to be averaged. For the calculation of the curvatures, the displacements at the reflector along the vertical edges of the walls are used (Figure 7).

### 2.4. Loading Protocols

The loading protocols of the walls are shown in Figure 8. Up to three load stages were applied: LS 1 = 0.75∙F_y_; LS 2 = δ_act_(0.75∙F_y_)·4/3·2; LS 3 = δ_act_(0.75∙F_y_)·4/3·3, where F_y_ is the yield strength and δ_act_ is the hydraulic cylinder displacement. In each load stage, two cycles were driven, cycle X-1 and cycle X-2. In the first cycle, the actuator was stopped at the maximum north and south deflection to track and measure cracks and take pictures. The stop is shown as a horizontal line in the loading protocols. The number of load stages differs between the experiments. For NW 1 and NW 2, three load stages are used for the analysis. In the fourth load stage, the force dropped below 80 percent and the analysis is stopped. In the NW 3 experiment, failure occurred already in load stage two.

## 3. Results

Investigation results on the three shear walls are presented below; load application was defined by load protocols. For displacement control (cycles 2 and 3), cylinder displacement was used. Clamping stiffness was not the same in north and south directions. For a southerly horizontal force, prestressing bars stretch so that support points were softer than for a northerly horizontal force. There, the force could be applied directly to the steel structure and, as a result, forces in the south direction were smaller than in the north direction for the same hydraulic cylinder deflection, as seen in Section 3.1, Section 3.2 and Section 3.3. A completely symmetrical load application could not be achieved; but, for the author, it was more important that clamping could be applied in the basement without additional vertical constraints. 

### 3.1. NW 1

#### 3.1.1. Crack Patterns and Crack Opening

Crack patterns in the three loading stages are shown in Figure 9, where for the cantilever and clamping part of the wall, separate pictures were taken. Below the first steel yielding in load stage 1 (0.75·F_y_ in cycle 1-1), horizontal cracks from flexure occurred, with some extending beyond the central wall axis. They were slightly influenced by the reinforcement grid, which caused weakening under tension at the reinforcement position. The crack width in load stage 1 (cycle 1-1) is up to about 0.6 mm, as Figure 10 (top) shows.

In load stage 2 (cycle 2-1), additional cracks occurred, and crack openings progressed. Additional horizontal (flexural cracks) emerged at the wall edge and ran, inclined, toward the neutral axis, about 0.1 m away from the wall edge. Towards the neutral axis, the flexural stress decreased, while the shear stress increased, which led to the inclination of the cracks; crack widths were then up to 2 mm. The crack spacing was approximately 200 mm. The reinforcement ratio was just large enough for the occurrence of a continuous crack distribution and not only of individual cracks. Crack spacing was somewhat influenced by the horizontal reinforcement.

In load stage 3 (cycle 3-1), the number of cracks remained constant with an increase of individual crack widths near the ceiling cutout (clamping horizon). At cycle 3-2, the wall was already in the softening branch; extensive plastic deformations occurred in the steel, and in the edges of the wall, first bars cracked.

#### 3.1.2. Load–Displacement Behaviour

Load–deformation behaviour is shown as a load–drift diagram, using the advantage that drift from the lower clamped part can be added directly to the upper cantilevered part to see overall behaviour. Displacements were measured with displacement transducers (see Figure 3 and Figure 4). As Figure 11 (top) shows, cantilever wall drift in cycle 2-1 was about 0.4 to 0.6 percent; clamped part drift in cycle 2-1 was about 0.25 to 0.3 percent. With a geometrically similar distribution of the curvatures between the clamped and cantilever part (see also the curvature drawing along the stiffening wall in Figure 1), upper wall section drift would be twice as large as lower section drift because the cantilevered wall section was twice as long as the clamped wall section. The measurement showed a similar result to that. This means that the clamped part behaves geometrically similar to the cantilever wall part. The top displacement is then 1.5 times greater compared to a pure cantilever displacement. However, in cycle 3.1, the deformation localises in one singe crack and the geometric similarity between cantilevered and clamped part no longer exists. Sliding displacements do not occur, as Figure 11 (right) shows.

In Figure 12, curvatures and centre strains are given. Both were calculated from measured displacements along the outer northern and southern measuring points of the optical measurement. The curvatures’ course correlated to the crack width course, as seen in Figure 10 and Figure 12. This makes sense, since large crack openings result in large steel strains, and thus curvatures also expand. The peaks in Figure 10 (left) can also be seen in Figure 12 (top left). 

In Figure 12 (bottom left), the horizontal displacements are shown, measured with optical measurement points along the wall’s centre axis. Therefore, displacement of the lowest point and the point 0.4 m above the ceiling axis was set to zero, which allows one to subtract the rigid body rotation with sufficient accurate approach. To use the displacements above the ceiling cutout for the rotation correction also allows one to make sliding shear displacements visible. In Figure 13 (bottom right), the (very small) difference between the measured and calculated displacements from the curvature distribution (Figure 12 (top left)) is shown. This means that horizontal displacements resulted almost exclusively from flexural curvatures, including the tension shift from inclined cracking due to interaction of flexure and shear. No sliding displacements occurred. 

### 3.2. NW 2

#### 3.2.1. Crack Patterns and Crack Opening

The three loading stages’ crack patterns are shown in Figure 13. Already, below the steel yielding in load stage 1 (0.75·F_y_ in cycle 1-1) inclined cracks formed. Near the boundary elements (3 × 2∅16), cracks were mostly horizontal, and the number of cracks was about twice that in the middle of the wall (secondary cracking). The part of the wall with inclined cracks was significantly larger compared to NW 1. That is because the shear force was over twice that of NW 1 (300 kN/140 kN = 2.1 in cycle 1). The cracks extended up to a height of about 60 percent of the wall height, with crack widths up to 0.15 mm in the cantilever part. In the clamping part, the maximum crack width was 0.3 mm (see Figure 14 (top)).

In load stage 2 (cycle 2-1), several additional cracks occurred, and the crack opening progressed. Especially in the middle, the cracks lengthened, and new cracks appeared up to approximately 80 percent of the height of the cantilever or basement wall. Crack distance at the boundary elements was about 100 mm and was, to a certain extent, influenced by the horizontal reinforcement and stirrups in the boundary elements (see Figure 6 and Figure 14). Inclined cracks occurred with a spacing between 200 mm and 300 mm. In both the boundary and inner parts, a distributed crack pattern occurred. Crack spacing in the experiment remained in the range of theoretical crack spacing, and crack widths on the boundary element’s face side were up to 2 mm. 

In load stage 3 (cycle 3-1), the number of cracks increased minimally compared to cycle 2-1, and the crack width also increased only minimally (Figure 14). Large sliding deformation occurred under the ceiling (see Figure 15 middle right), while flexure deformation (including tension shift from shear) somewhat decreased compared to cycle 2-1 in the cantilever part (see Figure 15 top left).

#### 3.2.2. Load–Displacement Behaviour

Figure 15 shows the drift of the cantilever and clamping part of NW 2. For cycle 2-1, cantilever drift was about 1.1 percent to the north and 0.8 percent to the south. In the clamped part, drift was about 0.5 percent to the north and 0.8 percent to the south. Geometric similarity between cantilever and clamping part—as for NW 1/cycle 2-1—could not be seen clearly in NW 2. In load stage 3, sliding displacements dominated the load–deformation behaviour. Considering Figure 15 (top left), cantilever wall drift in load stage 3 was not greater than in load stage 2, even in some cases it was smaller. The increase in the total drift resulted only from basement zone sliding displacements, shown in Figure 15 (middle right). Especially in cycle 2-2, 3-1, and 3-2, large sliding drifts occurred and significantly influenced total drift in Figure 15 (bottom). 

In Figure 16 (top), the curvatures and the centre strains are shown. The curvatures correlate with crack widths from Figure 14, as already was seen for NW 1. The curvatures are used to calculate horizontal displacements from flexure and tension shift (interaction with shear). Figure 16 (bottom left) shows optically measured horizontal displacements in the centre wall axis; Figure 16 (bottom right) shows the difference between measured displacements and displacements, calculated from the curvature distributions (Figure 16 (top left)). As already explained for NW 1, the rigid body rotation is calculated out by setting the displacement on the bottom support axis and 0.4 m above the ceiling cutout to zero. Sliding shear displacements occur below the ceiling cutout (–0.4 m) and are marked in Figure 16 (bottom right). The first significant difference occurred in cycle 2-1 (deflection to the north) and grew in cycle 3-1. Sliding displacements became more and more pronounced while the loading was already in the descending branch. In cycle 3-1 north, the deformed shape of the wall shows significant sliding displacements. A shear force hinge formed under the basement ceiling slab where the wall moved toward the load. This displacement can be clearly seen in Figure 16 (bottom right).

### 3.3. NW 3

#### 3.3.1. Crack Patterns and Crack Opening

The crack patterns of the two load stages are shown after cycle 1 in Figure 17. At 0.75·F_y_ in cycle 1-1, only a few horizontal cracks occurred. In the inner section, the cracks were already a little inclined, unlike NW 1, which showed almost exclusively horizontal cracks. The shear force for NW 3 was 1.79 times greater than for NW 1 (250 kN/140 kN = 1.79) and the shear stress about 1.33 times greater. The crack widths (Figure 18 (top)) were in the range of up to 0.3 mm, and one large crack occurred with a width of 1.2 mm in the clamped part on the north side, as Figure 18 (top right) shows.

In load stage 2, several additional cracks occurred. The crack opening proceeded up to a maximum crack width of 3.5 mm. The crack spacing was about 200 mm (Figure 18 (bottom), like it was for NW 1. The crack patterns, from loading to the north and south, were not symmetric. It is suspected that the compression force from the weight of the hydraulic cylinder on the upper north side leads to less cracking in the cantilever part. In the clamping part, a tension force is introduced from the stilt, which leads to more cracking on the north than on the south side. Additionally, the dead load of the cantilever supports on the vertical support on the north side and the dead load of the clamped part hang on this support. These vertical forces are about of 10 percent of the chord forces, which influences especially the crack pattern between the cantilevered and clamped part of NW 3. The total displacement, which includes both parts, is rather decoupled from the division between top and bottom. For a deflection to the north, the crack behaviour resembled NW 1: long horizontal (flexural) cracks occurred, which are inclined in the vicinity of the neutral axis. Cracks are uniformly distributed and again somewhat influenced by the horizontal reinforcement.

#### 3.3.2. Load–Displacement Behaviour

Figure 19 shows the drift of the cantilever and clamping part, as well as total drift of NW 3. As already explained previously, behaviour for a deflection to the north was different from that to the south. The total drift south (Figure 19 (bottom left)) resulted mainly from cracking in the clamping part (Figure 19 (middle left)). The drift north resulted from cracking in the cantilever (Figure 19 (top left)) and clamping part (Figure 19 (middle left)). For cycle 2-1, the drift was 0.64 percent in the cantilever and 0.27 percent in the clamping part. The factor between both is about two, as for NW 1. This, in turn, would indicate that the upper and lower wall sections behave in a geometrically similar way.

Figure 19 (right) shows the sliding shear drifts. In cycle 2-1, sliding shear began in the clamped wall part and increased in cycle 2-2 (Figure 19 (middle right)). In cycle 2-2 north, the force already dropped to 2/3 of the maximum force. In the following driven cycle 2-2 south, the force dropped down to 25%. Several vertical rebars cracked. For this reason, the experiment was stopped after cycle 2-2.

In Figure 20, curvatures and centre strains calculated from the optical measurement are given. Crack widths and curvatures correlate with each other, as it was for NW 1 and NW 2 (see Figure 18 (top) and Figure 20 (top)). In Figure 20 (bottom right), the difference between the measured displacements and “curvature distribution” displacements are shown. For cycle 2-1, a small difference can be seen, which indicates the start of sliding shear displacements. Larger sliding shear displacements occurred later in cycle 2-2, which is not presented in the figure. Thus, up to cycle 2-1, the displacements resulted mainly from the flexural deformations complemented by the tension shift effect.

The envelopes of all walls are presented in Figure 19 (bottom right). The differences in the force–displacement behaviour can be seen there.

## 4. Load–Displacement Analysis of the Cantilever Part

A precise calculation of the load–displacement is of crucial importance to estimate the capacity of a shear wall to resist earthquake loadings. The flexural deformation provides the main part of the plastic capacity to dissipate energy. Displacements from shear or sliding occur suddenly and lead to brittle failure. Therefore, the focus is on the bending deformations of the cantilever part of the wall. The clamped part is not considered within this study because in squat walls, the bending behaviour is less dominant. Figure 21 (left) shows the fan-like cracking of the cantilevered wall schematically. On the right side, two curvature distributions are sketched: the continuous line applies for the distribution with tension shift and the dashed line for the distribution without tension shift, resp. for pure bending (Bernoulli). The discrepancy between both lines results from inclined cracking due to the interaction between flexure and shear. This is known as the tension shift effect, which has an important impact on the load–displacement curve. The yield point shifts upwards because the plastic strains spread over a greater height. The tension resultant T is on a higher level than the belonging compression resultant C. For pure bending, they would be at the same height. The impact of the tension shift effect on the load–displacement curve is analysed in the following.

In Figure 22 and Table 4, the measured curvatures are compared to the calculated curvatures from pure bending (Bernoulli). The stepped lines from the experiments are approximated by bilinear approaches, which are extended up to the ceiling cutout (exp). The experimental lines are compared to the bilinear calculations from the moment–curvature analysis for pure bending (Bernoulli). One can see that the differences between both curves are significant, which leads to significant differences in the load–displacement curves, as Figure 23 and Table 5 show. The height of the yield point above the clamping section results from two parts: first, from the slope of the hardening line in the moment–curvature curve and second, from the tension shift of the yield point due to inclined cracking. The schematic in Figure 21 (left) shows that the shift takes place over the length between the neutral axis and the tension resultant, z_T_. The corresponding shift height at the yield point, h_shift_, is the difference between the measured yield point and the calculated one from pure bending. The crack angle at the yield point is $\theta_{y}=\tan^{-1}z_{T}/h_{shift}$ and between 55° and 65°. The crack patterns in Figure 9, Figure 13 and Figure 17 show the fan-like cracking, which is partly straight and partly curved. In the sketch in Figure 21, straight lines are assumed, which is a simplified approximation of the experiments.

Neglecting the tension shift effect leads to too small maximum displacements. As Table 5 indicates, the factor between δ_flex_ and δ is from 0.36 to 0.73, which means that a calculation on a basis only on Bernoulli leads to only 36% to 73% of the measured displacements. The tension shift effect contributes enormously to the maximum displacement of a shear wall. A comparison in Table 5 of the drift from the calculation ϕ to the experiment ϕ_exp_ shows that the results for NW 1—cycle 2 and NW 3 are nearly equal. For NW 1—cycle 3 and NW 2, the calculated results are higher than the measured. The difference is up to 14%, which is still a good result for a bilinear approximation of the curvature distribution and the force–displacement curve.

As stated in Section 3, the main deformations result from the curvatures over the height. Shear distortions do not play a roll. In NW 2 and NW 3, a premature sliding failure occur in the clamping part. Up to this failure, it is sufficient to calculate the head displacement of a wall on the basis of a moment–curvature analysis for pure bending and add the tension shift effect in the curvature distribution over the height of the wall.

## 5. Conclusions

The presented study gives more insights into the deformation and failure behaviour of into-the-basement-clamped RC shear walls:

### 5.1. Crack Pattern

In the walls with slight longitudinal reinforcement (NW 1 and NW 3), the cracks ran horizontally from the edges to the middle of the wall. Towards the neutral axis, cracks run increasingly inclined since the shear stress increased compared to the bending stress. In the wall with more reinforcement in the boundary (NW 2), cracks were horizontal only in the edge zones. Between the edge zones, respectively, boundary zones, cracks were inclined nearly over the complete distance.

### 5.2. Crack Opening and Crack Spacing

All walls (NW 1, NW 2, and NW 3) showed distributed crack patterns with roughly constant spacing between the cracks. For the wall with more reinforcement (NW 2), the spacing was significantly smaller. Secondary cracks occurred in the reinforced boundary zones. For walls with smaller longitudinal reinforcement ratios, the deformations localised in a single crack in the loading softening branch. One crack opened strongly, while the other cracks did not change or closed slightly. In the wall with more reinforcement (NW 2), this localisation of the deformation in one single crack did not occur; deformations were distributed over several cracks.

### 5.3. Curvature Distribution over the Height

For walls with less reinforcement (NW 1 and NW 3), curvatures localised near the clamping zone. The tensile strain shifts (plastic spread) were smaller than for walls with more reinforcement in the edge zones (NW 2). More reinforcement in the edge zones resulted in more inclined cracking and, hence, in a greater shift of tensile strains and, respectively, curvatures. The wall with the longer length (NW 3) showed a slightly greater shift than the shorter wall (NW 1). Displacements from the curvatures were thus dependent on the reinforcement ratio, the distribution of reinforcement over the cross-section, and the wall aspect ratio H/L, as this study shows.

### 5.4. Load–Displacement Behaviour

Displacements at the top of a wall resulted from the upper floors’ cantilevered part and from the clamped part of the basement floor. If the curvature distributions of the two parts are affine to each other, drifts are divided according to the lengths, ϕ_up_/ϕ_base_ = l_up_/l_base_. The total drift is ϕ_tot_ = ϕ_up_ + ϕ_base_. For NW 1, NW 2, and NW 3, l_up_/l_base_ is about 2, so that about 1/3 of the total displacement would result from the basement wall and 2/3 from the cantilever wall. The experiments showed a tendency in this direction; however, for NW 2 and NW 1, cycle 3-1 behaves different. There, one localised single crack opened in the cantilever part, which yields to a deformation concentration in this part.

A calculation of the head displacement from the curvature distribution over the total height of the wall (cantilever + clamping part) leads to good results. In a first step, the curvatures from a cross-sectional analysis for pure bending (Bernoulli) can be used. The tension shift effect must then be added in the curvature distribution over the wall height so that the interaction with the shear force is added. This means that the tension shift effect has a greater impact on walls with smaller height to length ratio. The influence decreases with the increase in the slenderness of a wall.

If the tension shift effect is neglected, maximum calculated displacements are only 36% to 73% of the measured one. This must be considered, especially when larger shear forces occur, e.g., due to more reinforcement in the edge element zones.

### 5.5. Sliding Failure

In two experiments, sliding shear occurred along the construction joint between the basement wall and the ceiling. It must be considered for walls with aspect ratios of approximately H/L = 1 and less.

### 5.6. Recommendations for Further Research

The author proposes two topics that could be investigated further on the basis of the study presented:
-The tension shift effect has a major influence on the force–displacement curve. Further, shear wall experiments from the literature could be evaluated to identify the influence of the tension shift effect on the plastic hinge length. One should be able to specify the individual effects from flexure (Bernoulli) and tension shift separately because they vary with the slenderness of a wall separately.-As mentioned in the publication, the sliding shear behaviour needs to be evaluated in more detail. For example, how is the stress and deformation state in a wall before sliding shear occurs? What are the impact factors on the sliding shear resistance for into-the-basement-clamped RC shear walls.

## Figures and Tables

**Figure 1 materials-17-02267-f001:**
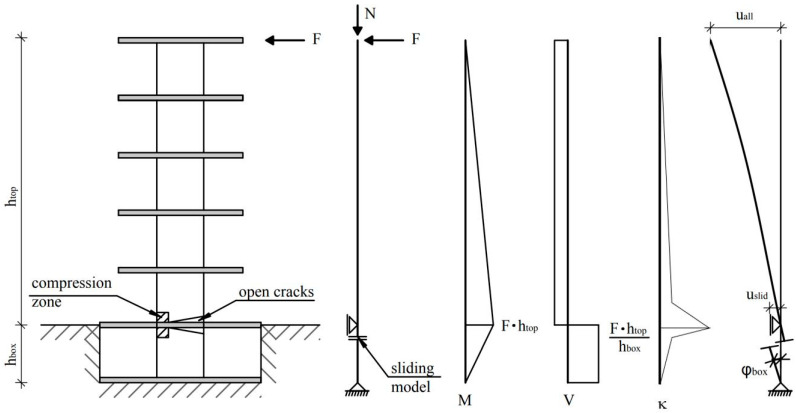
Shear wall clamped in the basement; internal forces; curvatures; deformations including sliding shear failure [17].

**Figure 2 materials-17-02267-f002:**
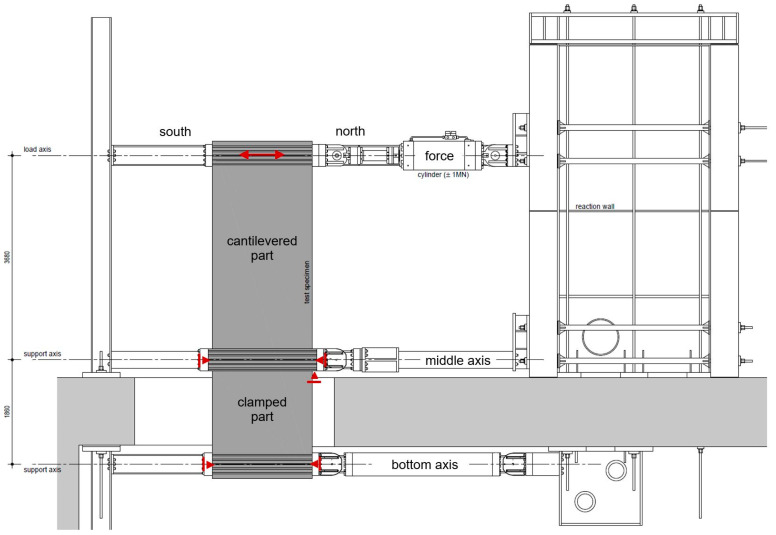
Front view of the experimental setup and static system in red.

**Figure 3 materials-17-02267-f003:**
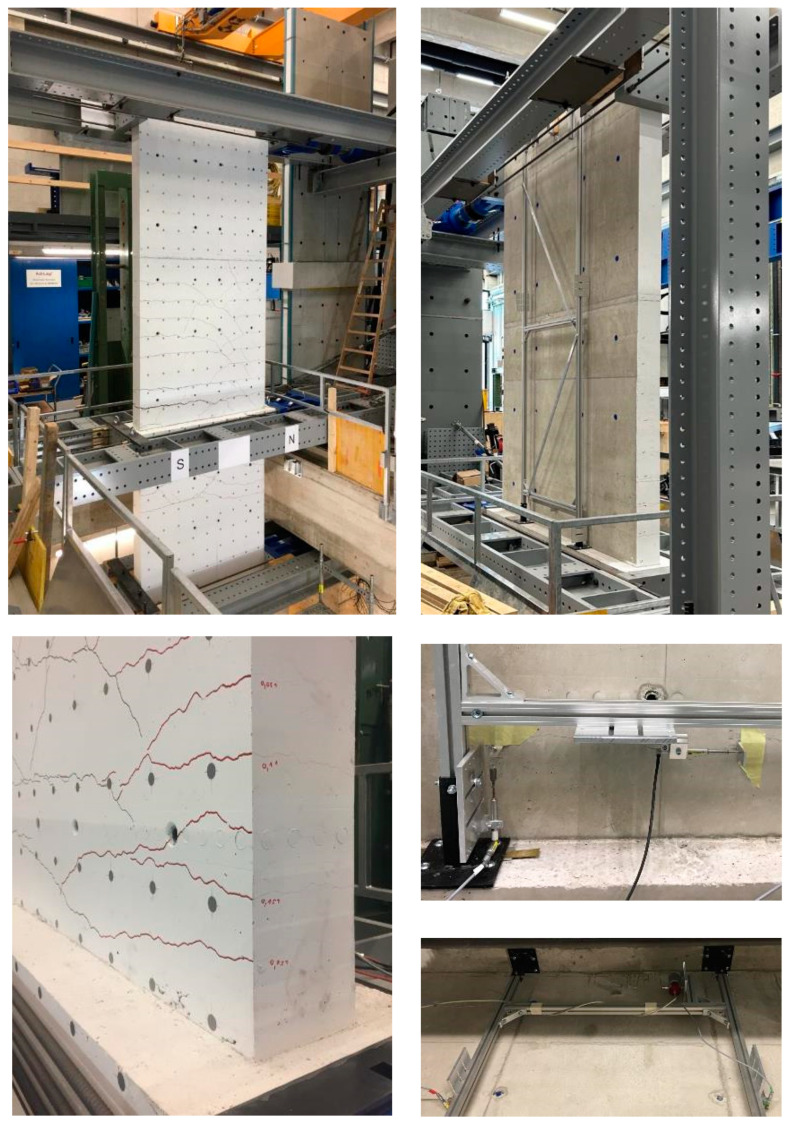
Shear wall test: front side (**top left**); back side with measuring frame mounted on the ceiling cutout (**top right**); crack widths on the side (**bottom left**); displacement transducers above and below the ceiling cutout (**bottom right**).

**Figure 4 materials-17-02267-f004:**
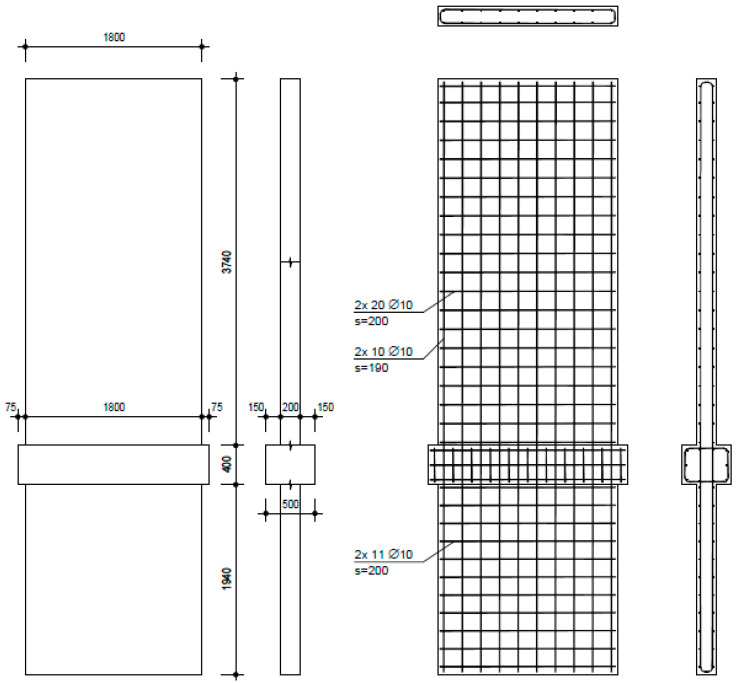
NW 1 formwork and reinforcement plans (dimensions in mm).

**Figure 5 materials-17-02267-f005:**
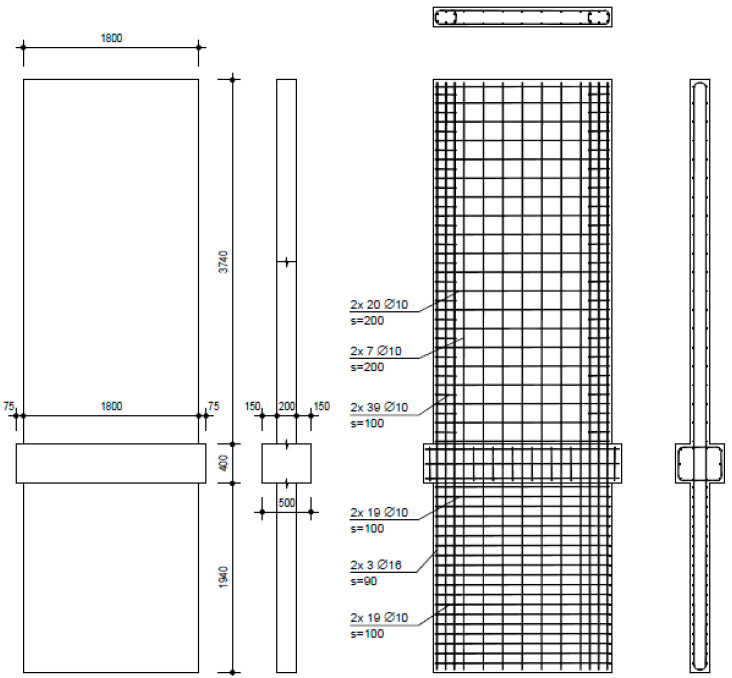
NW 2 formwork and reinforcement plans (dimensions in mm).

**Figure 6 materials-17-02267-f006:**
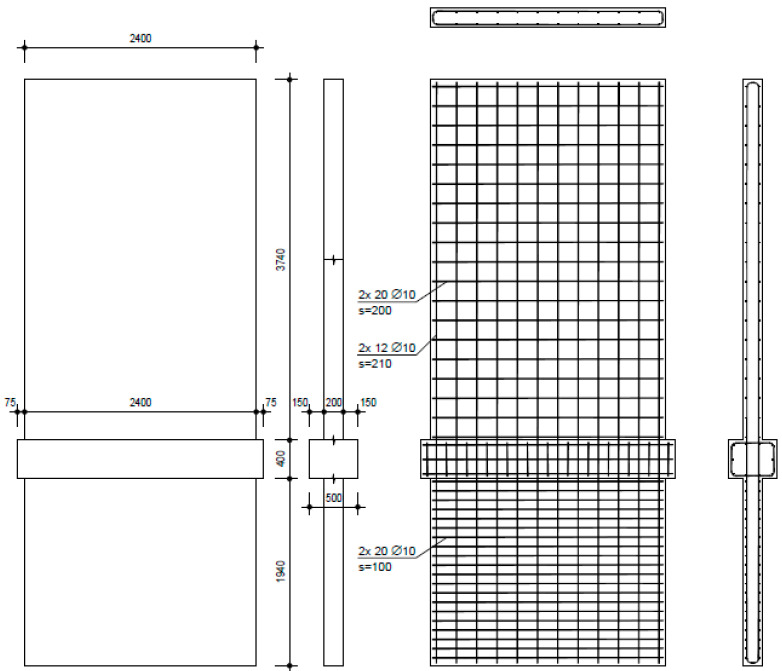
NW 3 formwork and reinforcement plans (dimensions in mm).

**Figure 7 materials-17-02267-f007:**
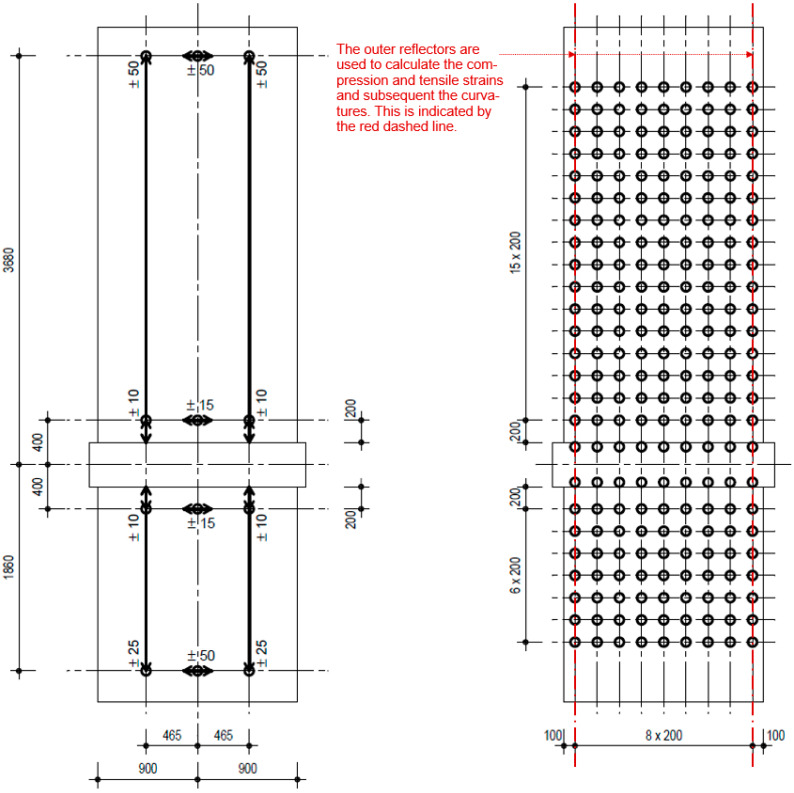
Measurement points of the displacement transducers (**left**); reflector grid of the optical measurement (**right**).

**Figure 8 materials-17-02267-f008:**
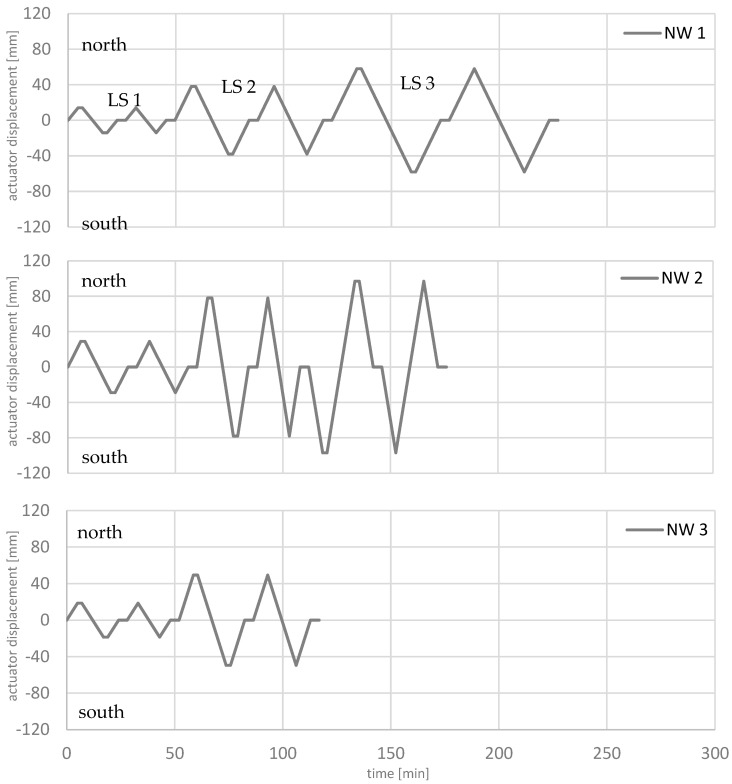
Loading protocols of the walls.

**Figure 9 materials-17-02267-f009:**
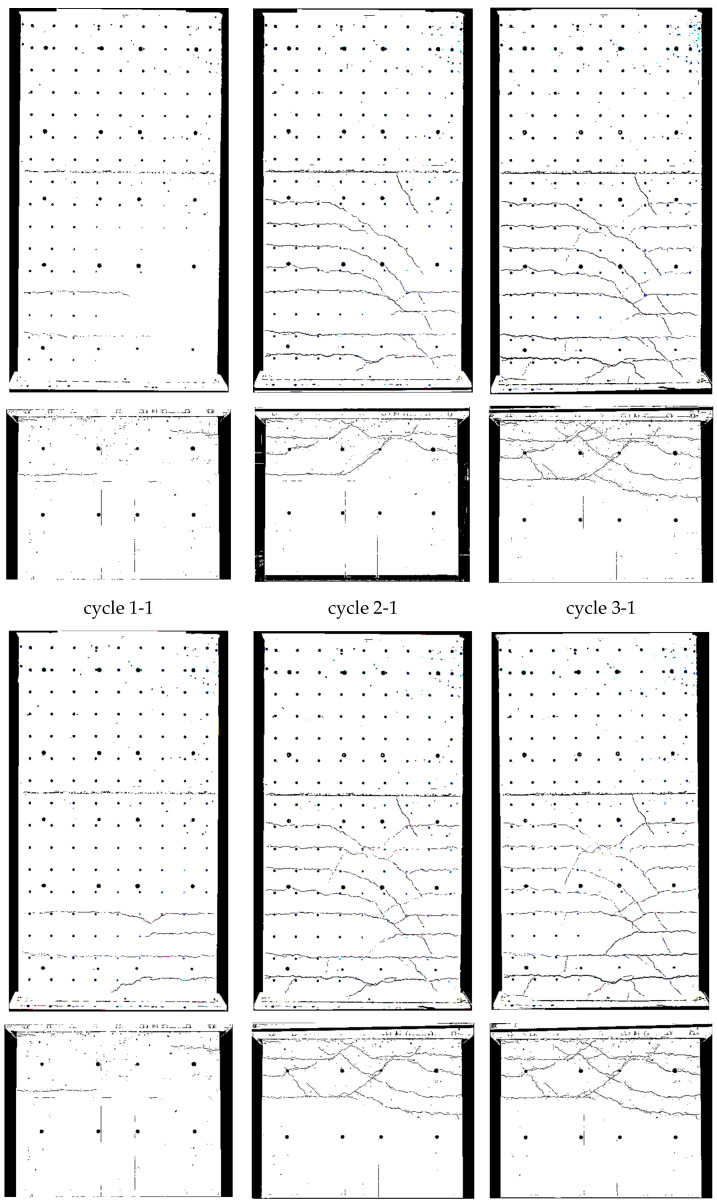
NW 1 crack pattern: deflection to the north (**top**) and to the south (**bottom**).

**Figure 10 materials-17-02267-f010:**
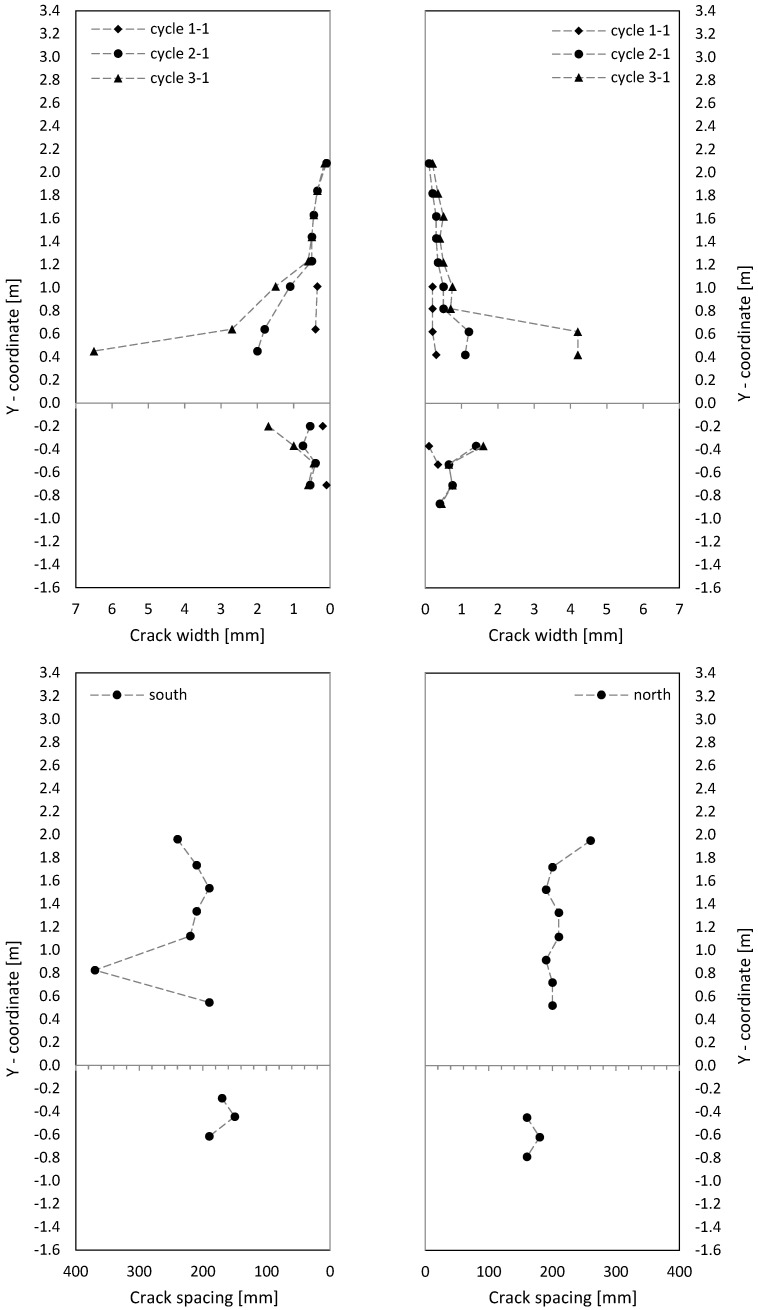
Crack width (**top**) and crack spacing (**bottom**) on the side of NW 1.

**Figure 11 materials-17-02267-f011:**
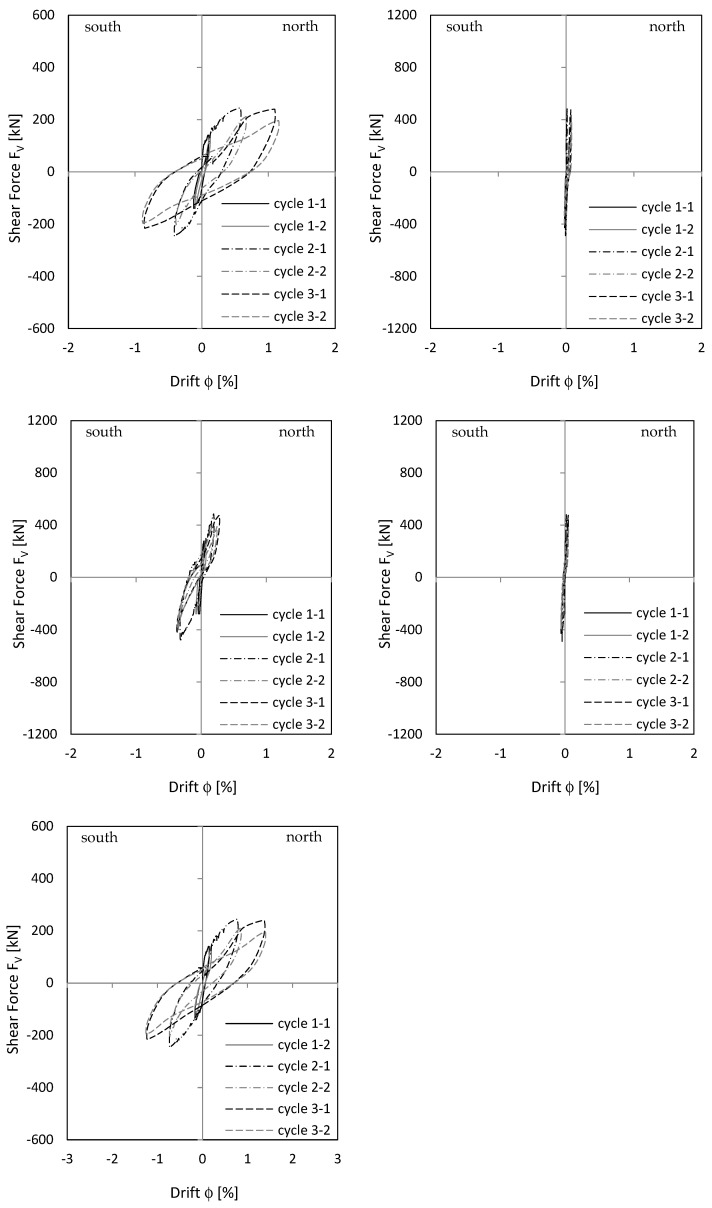
NW 1 load—drift relationship: cantilever part (**top left**), clamping part (**middle left**), total = cantilever + clamping part (**bottom left**), sliding cantilever part (**top right**), sliding clamping part (**middle right**).

**Figure 12 materials-17-02267-f012:**
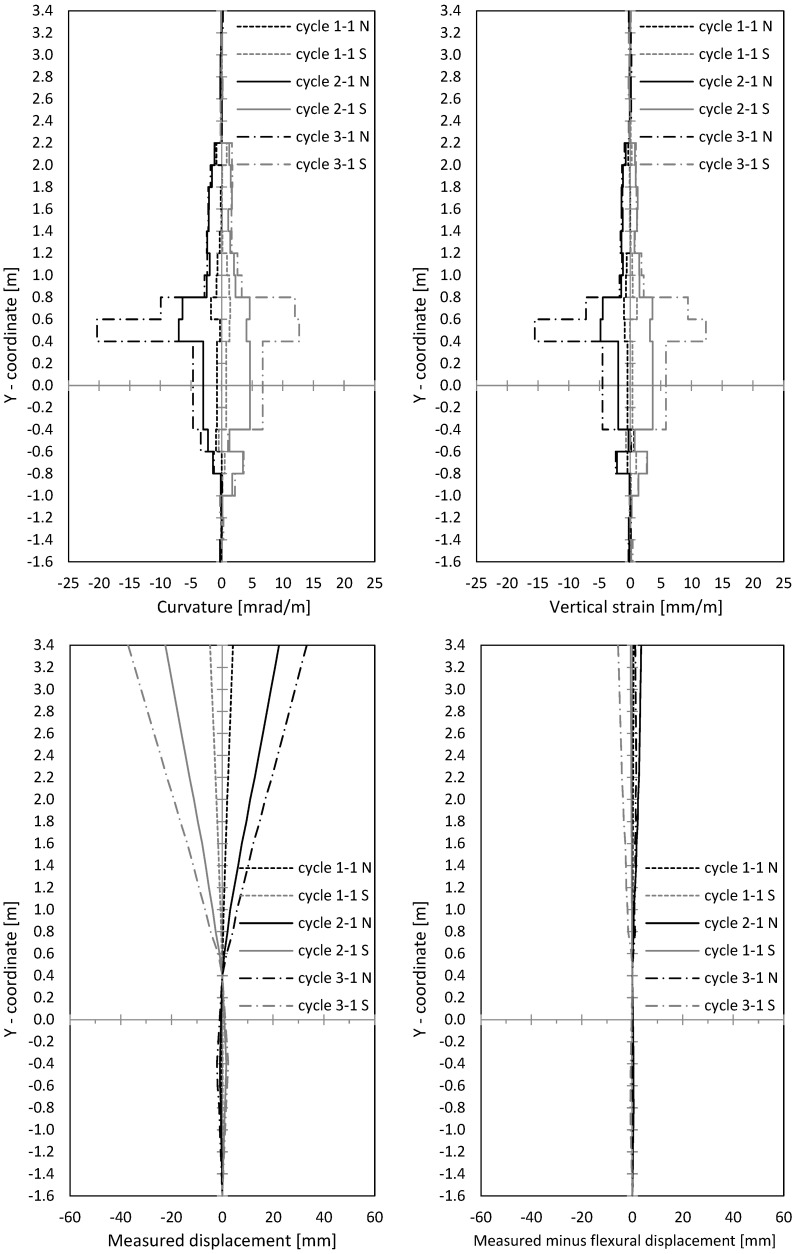
NW 1: Curvatures and vertical strains over the height (**top**); measured horizontal displacements and measured displacements minus displacements calculated from the measured curvatures (**bottom**).

**Figure 13 materials-17-02267-f013:**
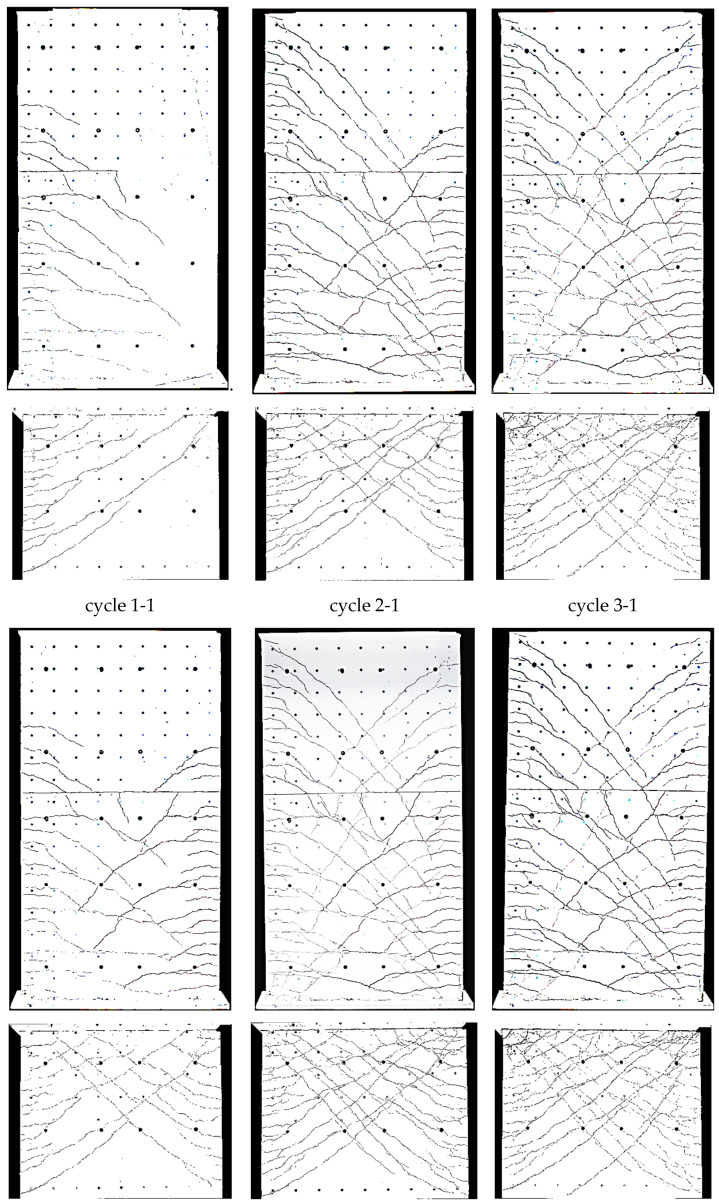
NW 2 crack pattern: deflection to the north (**top**) and to the south (**bottom**).

**Figure 14 materials-17-02267-f014:**
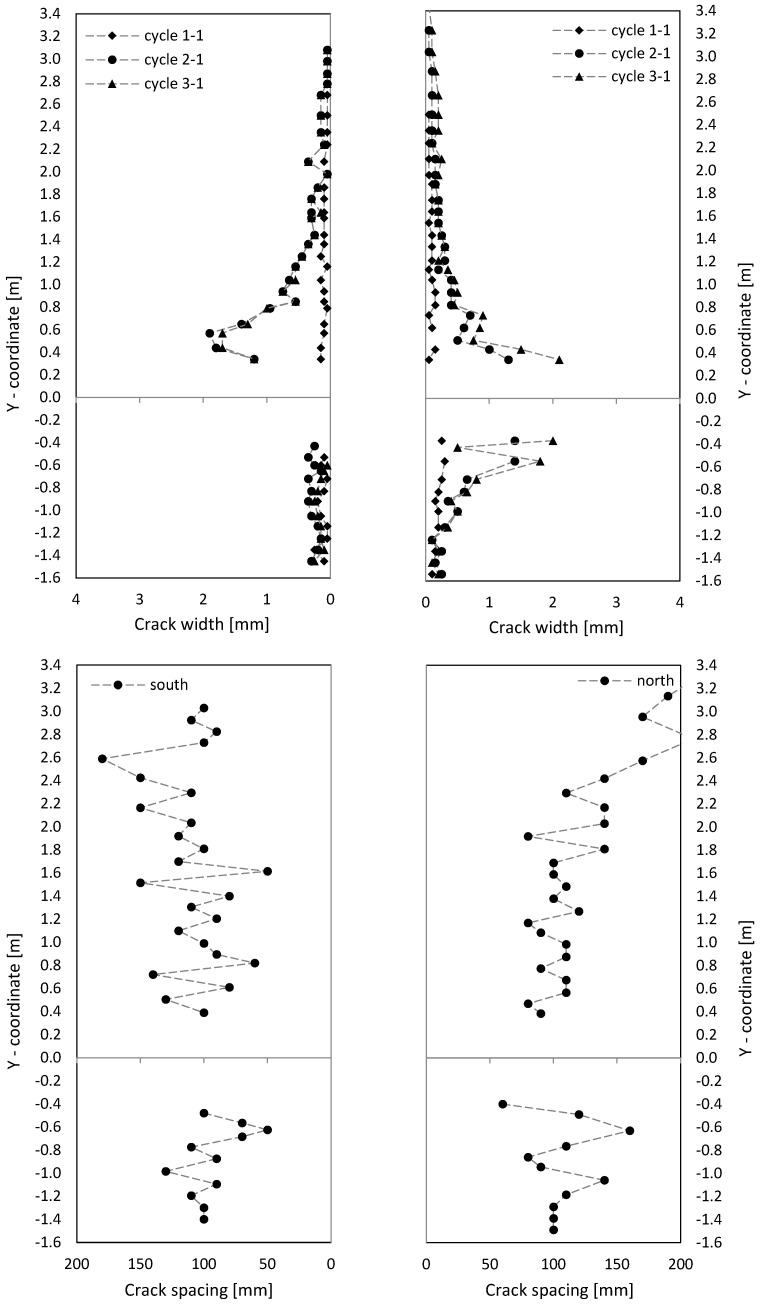
Crack width (**top**) and crack distance (**bottom**) on the side of NW 2.

**Figure 15 materials-17-02267-f015:**
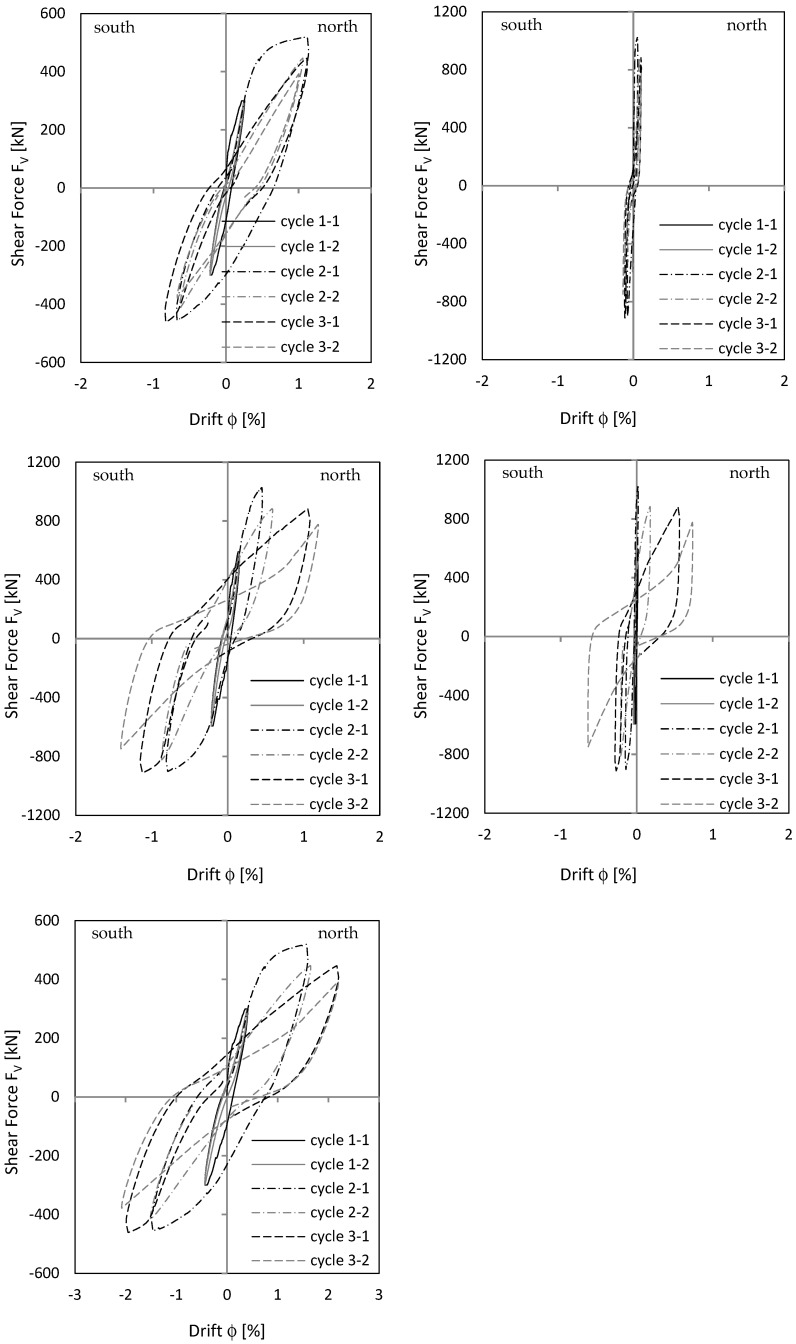
NW 2 load—drift relationship: cantilever part (**top left**), clamping part (**middle left**), total = cantilever + clamping part (**bottom left**), sliding cantilever part (**top right**), sliding clamping part (**middle right**).

**Figure 16 materials-17-02267-f016:**
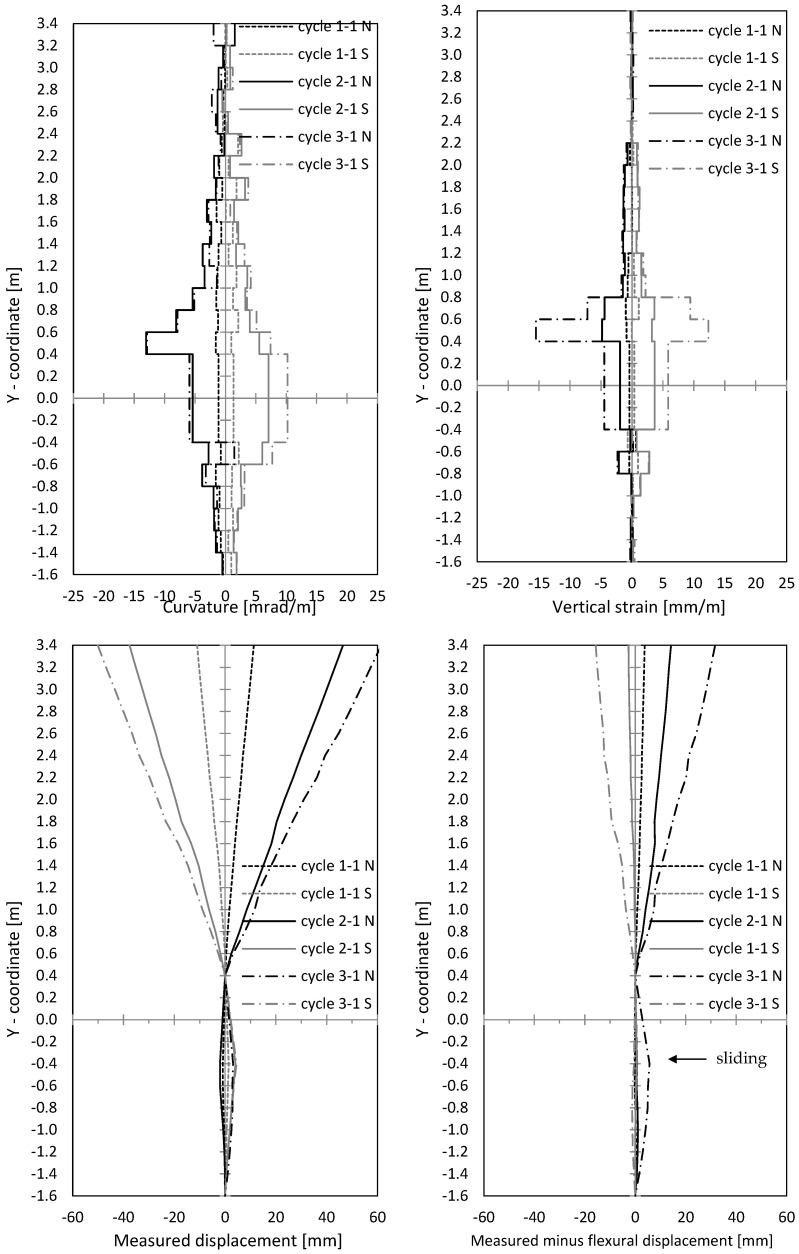
NW 2: curvatures and vertical strains over the height (**top**); measured horizontal displacements and measured displacements minus displacements calculated from the measured curvatures (**bottom**).

**Figure 17 materials-17-02267-f017:**
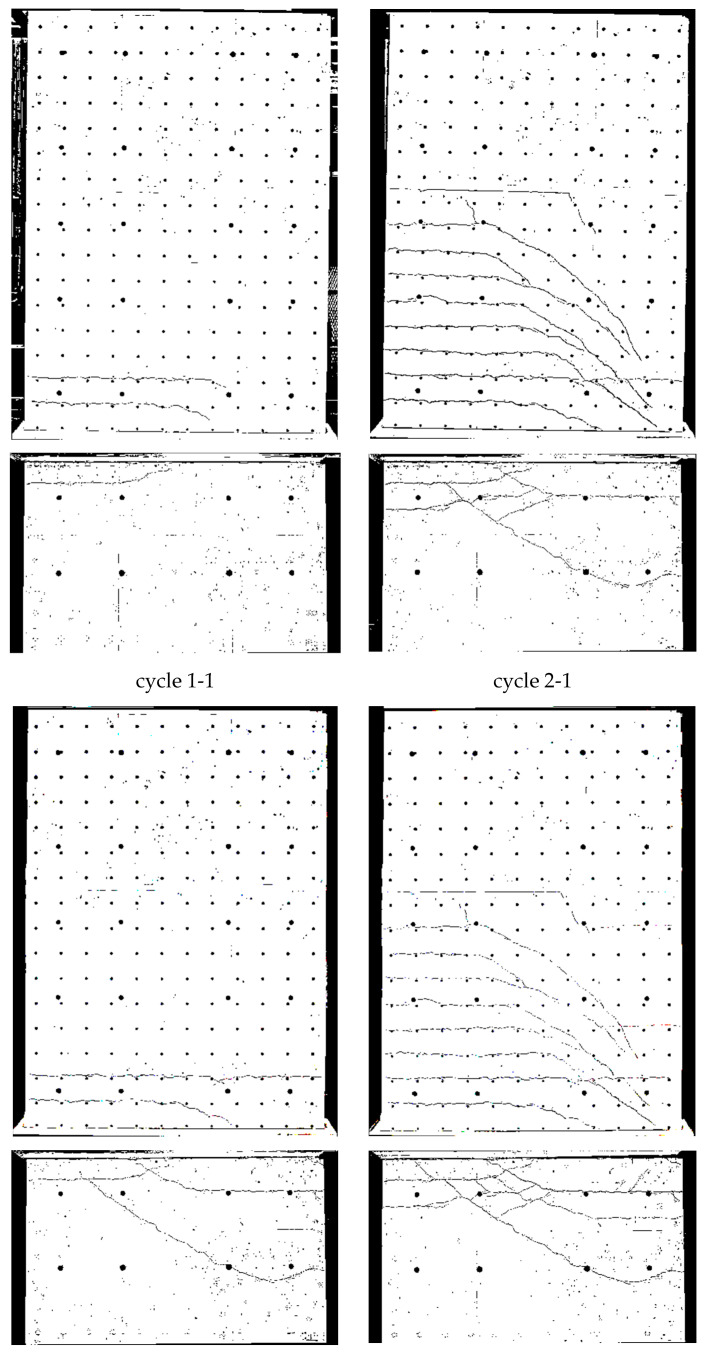
NW 3 crack pattern: deflection to the north (**top**) and to the south (**bottom**).

**Figure 18 materials-17-02267-f018:**
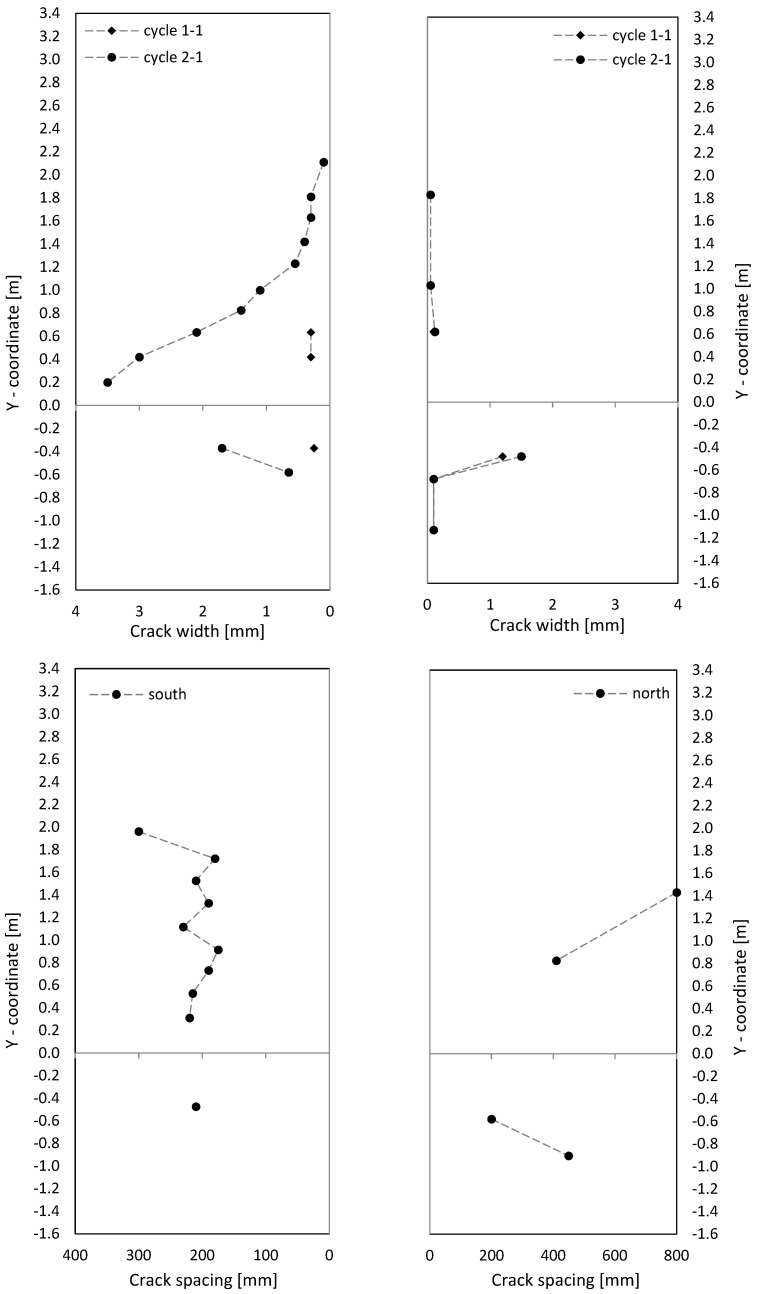
NW 3 crack width (**top**) and crack spacing (**bottom**) on the side.

**Figure 19 materials-17-02267-f019:**
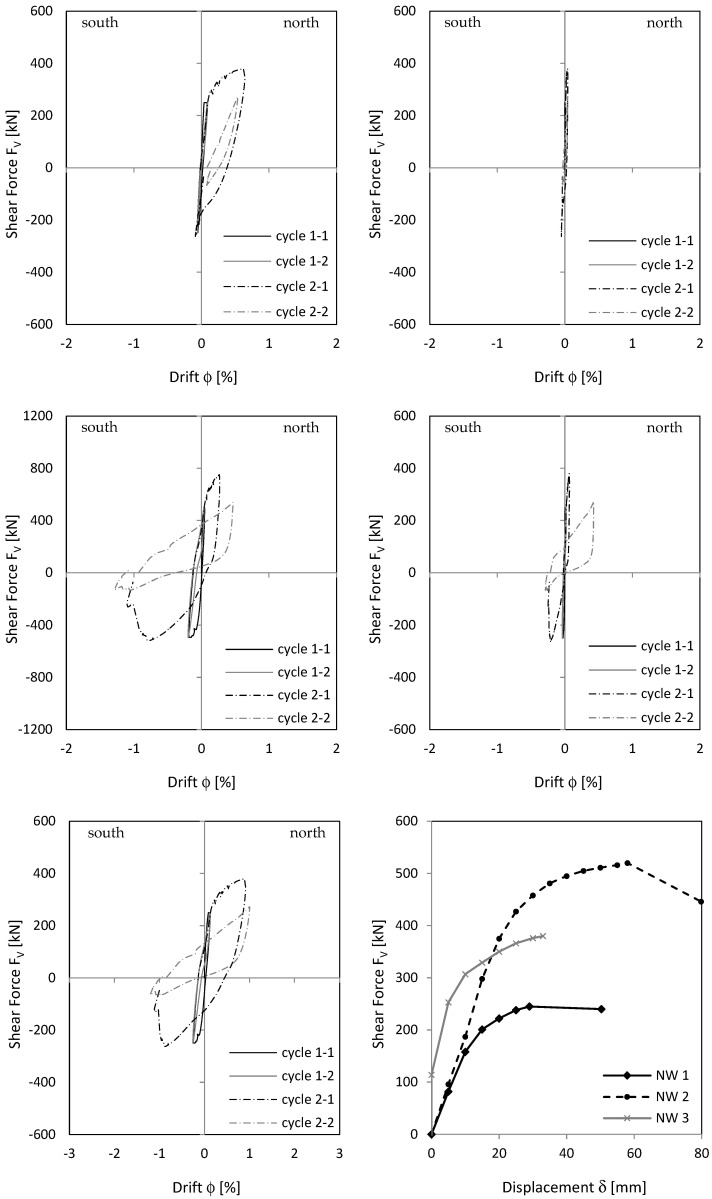
NW 3 load–drift relationship: cantilever part (**top left**), clamping part (**middle left**), total = cantilever + clamping part (**bottom left**), sliding cantilever part (**top right**), sliding clamping part (**middle right**). Envelope of NW 1, NW 2, and NW 3 (**bottom right**).

**Figure 20 materials-17-02267-f020:**
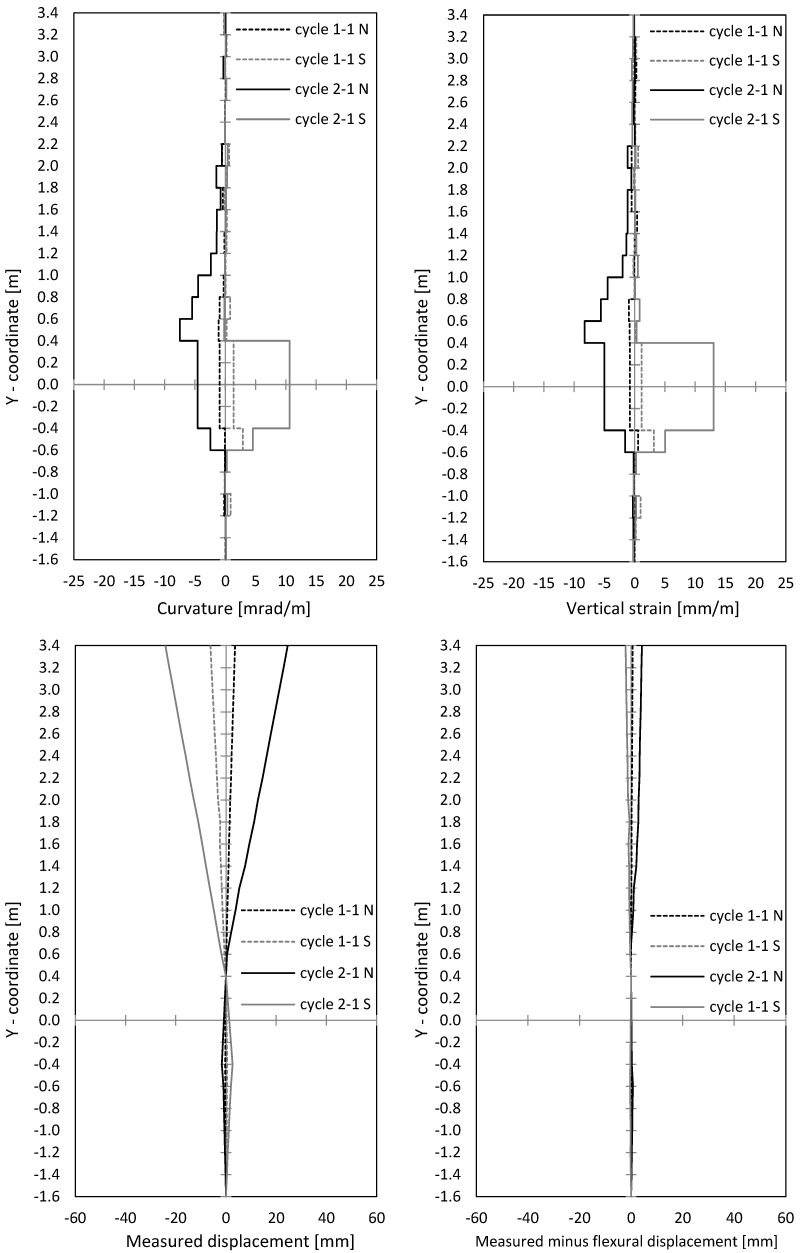
NW 3: curvatures and vertical strains over the height (**top**); measured horizontal displacements and measured displacements minus displacements calculated from the measured curvatures (**bottom**).

**Figure 21 materials-17-02267-f021:**
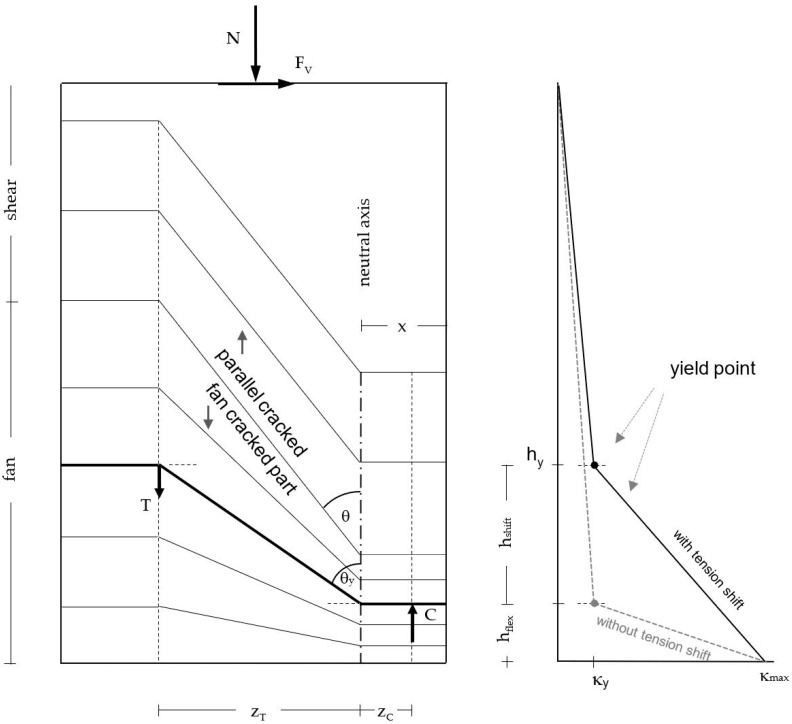
Shear wall: schematic crack pattern (**left**); bilinear curvature distribution with and without tension shift (**right**).

**Figure 22 materials-17-02267-f022:**
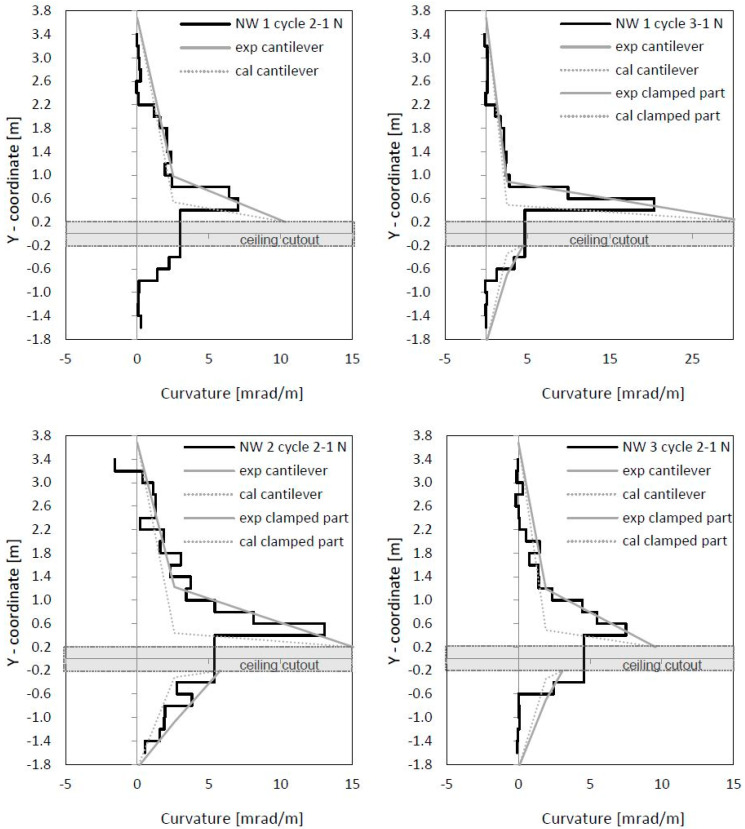
Curvature distribution over the height: experiment, bilinear approximation, and calculation from pure bending.

**Figure 23 materials-17-02267-f023:**
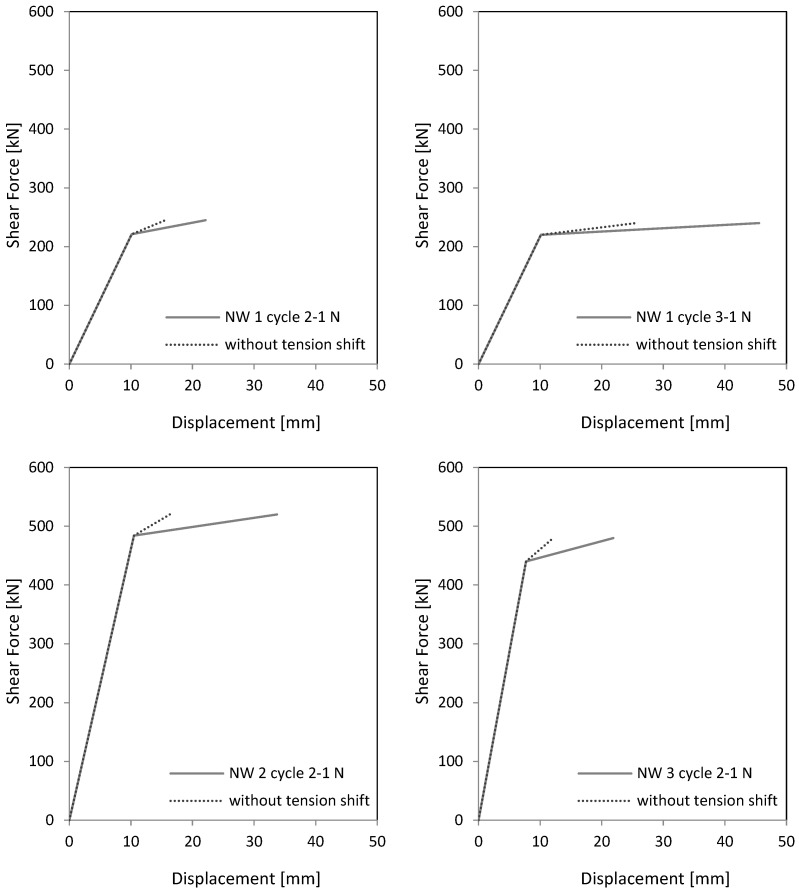
Bilinear load–displacement curves calculated from the bilinear curvatures in Figure 22.

**Table 1 materials-17-02267-t001:** Shear wall geometry and reinforcement.

Wall	h_up_	h_base_	t	l_b1/2_	l_web_	ρ_b1/2_	ρ_web_	ρ_V_up_	ρ_V_base_
	[m]	[m]	[m]	[m]	[m]	[%]	[%]	[%]	[%]
NW 1	3.68	1.86	0.20	0	1.80	-	0.44	0.39	0.39
NW 2	3.68	1.86	0.20	0.27	1.26	2.23	0.44	0.39	0.79
NW 3	3.68	1.86	0.20	0	2.40	-	0.39	0.39	0.79

h_up_ = height of cantilever wall from the middle support to the upper load axis; h_base_ = height of basement wall from the lower to the middle support axis; t = wall thickness; l_b_ = length of reinforced boundaries; l_web_ = length of web without boundaries; ρ_b_ = boundary reinforcement ratio; ρ_web_ = web reinforcement ratio; ρ_V_up_ = cantilever wall transverse reinforcement ratio; ρ_V_base_ = basement wall transverse reinforcement ratio.

**Table 2 materials-17-02267-t002:** Concrete material properties on testing date.

Wall	f_c_cube_base_	f_c_cube_up_	f_st_base_	f_st_up_
	[MPa]	[MPa]	[MPa]	[MPa]
NW 1	42.2	38.3	3.1	2.7
NW 2	52.6	35.9	4.1	3.2
NW 3	44.1	39.8	3.3	2.9

f_c_cube_base_ = basement wall concrete cube compressive strength; f_c_cube_up_ = concrete cube compressive strength in cantilever wall adjacent to the ceiling; f_st_base_ = basement wall concrete splitting tension strength; f_st_up_ = concrete splitting tension strength in cantilever wall adjacent to the ceiling.

**Table 3 materials-17-02267-t003:** Reinforcement material properties.

∅	E_s_	f_sy_	f_su_	ε_y_	ε_u_
	[GPa]	[MPa]	[MPa]	[mm/m]	[mm/m]
10	202	601	635	3.0	65.3
16	186	606	658	3.3	119

E_s_ = steel Young’s modulus; f_sy_ = steel yield strength; f_su_ = steel tensile strength; ε_y_ = steel yield strain; ε_u_ = steel ultimate strain.

**Table 4 materials-17-02267-t004:** Parameters for the determination of the shear crack angle in the fan at the yield point.

Wall	Cycle	l_w_	h_w_	κ_y_	κ_max_	h_flex_	h_y_	h_shift_	x	z	z_T_	θ_y_
	[-]	[m]	[m]	[mrad/m]	[mrad/m]	[m]	[m]	[m]	[m]	[m]	[m]	[°]
NW 1	2-1 N	1.80	3.48	2.51	10.3	0.34	0.78	0.44	0.16	0.98	0.88	63
NW 1	3-1 N	1.80	3.48	2.51	32.0	0.29	0.69	0.4	0.13	0.94	0.87	65
NW 2	2-1 N	1.80	3.48	2.60	15.1	0.24	1.03	0.79	0.18	1.24	1.14	55
NW 3	2-1 N	2.40	3.48	1.91	9.6	0.29	0.99	0.7	0.19	1.29	1.17	59

C = compression resultant; T = tension resultant; l_w_ = wall length; h_w_ = wall height from the top edge of the ceiling cutout to the upper load axis; κ_y_ = yield curvature from the bilinear fit on the moment–curvature relationship; κ_max_ = maximum curvature at the ceiling cutout obtained from the fit in Figure 22 on the measured curvatures in the experiments; h_flex_ = height of the yield point calculated out of the moment–curvature curve (B = Bernoulli for pure bending without tension shift); h_y_ = height of the yield point of the bilinear approximation of the curvatures in the experiments in Figure 22 (with tension shift); h_shift_ = h_y_ − h_flex_ = length of the tension shift at the yield point due to inclined cracking from interaction between bending and shear; x = compression zone height; z = lever arm; z_T_ = distance from neutral axis to the tension resultant T; θ_y_ = tan^−1^(z_T_/h_shift_) = shear crack angle in the fan at the yield point; θ = shear crack angle (outside of the fan). For the designations see also Figure 21.

**Table 5 materials-17-02267-t005:** Displacements without and with tension shift and comparison of the drifts, calculated from the approx. curvature to the measured drifts.

Wall	Cycle	δ_flex_	δ	ϕ	ϕ_exp_	Figure
		[mm]	[mm]	[%]	[%]	(Experiment)
NW 1	2-1 N	16	22	0.60	0.58	11 top left
NW 1	3-1 N	26	46	1.25	1.10	11 top left
NW 2	2-1 N	16	44	1.20	1.12	15 top left
NW 3	2-1 N	12	22	0.60	0.61	19 top left

δ_flex_ = displacement of the cantilever, calculated from pure bending curvatures (Bernoulli) out of Figure 22 = displacement without tension shift; δ = displacement of the cantilever, calculated from the measured curvatures out of Figure 22 = displacement with tension shift; ϕ = drift from δ = drift from the measured curvatures; ϕ_exp_ = drift measured in the experiments in Figure 11, Figure 15 and Figure 19 (top left).

## Data Availability

Data are contained within the article.

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
