# Peer review of "Flexural and Shear Deformation of Basement-Clamped Reinforced Concrete Shear Walls"

_materials, 2024, doi:10.3390/ma17102267_

Round 1
Reviewer 1 Report
Comments and Suggestions for Authors
The authors presents three shear wall experiments with heights up to six meters. Measured were the force-displacement curve, the curvature distribution over the height, the crack pattern, the crack opening and spacing. The research content of the manuscript clearly conforms to the scope of the journal. The research content of the manuscript will interest the readers. However, it needs some revisions according to some comments as below:
1- The section of introduction is too simple. It is recommended that authors provide a detailed overview of reinforced concrete shear wall tests and walls' clamping parts.
2- Wall 3 seems not to be displayed in the manuscript, what’ the reason?
3- The format of references is relatively chaotic and the quantity is relatively small. Please carefully check and maintain consistency.
4- The resolution of figures is needed to be improved by the authors, it is recommended to use vector image format.
5-The following literature maybe useful for improving your quality:
â‘ Shear mechanical responses of sandstone exposed to high temperature under constant normal stiffness boundary conditions. Geomechanics and Geophysics for Geo-Energy and Geo-Resources, 2021, 7:35.
â‘¡ Recognition of shear and tension signals based on acoustic emission parameters and waveform using machine learning methods. International Journal of Rock Mechanics and Mining Sciences, 2023, 171: 105578
Comments on the Quality of English LanguageThe authors presents three shear wall experiments with heights up to six meters. Measured were the force-displacement curve, the curvature distribution over the height, the crack pattern, the crack opening and spacing. The research content of the manuscript clearly conforms to the scope of the journal. The research content of the manuscript will interest the readers. However, it needs some revisions according to some comments as below:
1- The section of introduction is too simple. It is recommended that authors provide a detailed overview of reinforced concrete shear wall tests and walls' clamping parts.
2- Wall 3 seems not to be displayed in the manuscript, what’ the reason?
3- The format of references is relatively chaotic and the quantity is relatively small. Please carefully check and maintain consistency.
4- The resolution of figures is needed to be improved by the authors, it is recommended to use vector image format.
5-The following literature maybe useful for improving your quality:
â‘ Shear mechanical responses of sandstone exposed to high temperature under constant normal stiffness boundary conditions. Geomechanics and Geophysics for Geo-Energy and Geo-Resources, 2021, 7:35.
â‘¡ Recognition of shear and tension signals based on acoustic emission parameters and waveform using machine learning methods. International Journal of Rock Mechanics and Mining Sciences, 2023, 171: 105578
Author Response
Dear Reviewer
Thank you very much for the helpful review. I have change several things. Please find attached my replies.
Kind regards, Harald Schuler

Reviewer 2 Report
Comments and Suggestions for Authors
The work is of an experimental nature and concerns the behavior of stiffening concrete walls. The authors analyzed a key fragment of the stiffening wall on the lowest floor above the basement floor at the point of connection with the ceiling of the floor above the basement. Three models of walls made on a scale of 1:1.5 with different reinforcement were examined. All models were subjected to cyclic loading with increasing amplitude. The following relationships were analyzed: load, displacement, changes in wall curvature using classic LVDT transducers and the DIC system. The scratches and scratch widths were imaged manually. Empirical dependencies between the tested and analyzed parameters were demonstrated. I evaluate the work positively from a practical and cognitive point of view. There are some issues that need to be supplemented or clarified to improve the quality of the manuscript.
1. Abstract. I propose adding research results in a synthetic way, not only general results.
1. Chapter 1. The introduction is very modest. It is proposed to expand the analysis by providing the results of analyzes by other authors, and, above all, to emphasize the purpose of the research and the contribution to the disciplines of Civil Engineering.
2. Chapter 2.1. In addition to the description of the test stand, a static diagram of the tested walls with the location of the supports used and the location of the loads should be shown. The issue of the uncertainty of vertical loads acting on the wall also requires clarification.
3. Chapter 2.2. Basically, there is no measurement of the shear strain angles, which would help find the correlation between shear force and shear strain. It is also necessary to provide how the curvature was calculated in individual sections.
4. Chapter 3. At this point, I suggest adding an addition on how and why the values of horizontal wall loads were different.
5. Chapters 3.1-3.4. In each model, I propose showing where the original cracks occurred and in which load cycle. I suggest also showing the envelope of the load-displacement relationship.
6. Chapter 4. In addition to flexural deformations, there are also dimensional deformations in the wall, why this aspect was omitted. Of course, the flexural analysis is only valid for flexural deformations caused by the bending moment. There is no information about the dependencies according to which the displacements were calculated taking into account changes in the wall curvature (how the wall curvature was integrated.)
7. In Chapter 5, I propose to add directions for further work.
Author Response
Dear Reviewer
Thank you for the helpful review. I have several things changed.
Please find attached my replies.
Kind regards, Harald Schuler

Reviewer 3 Report
Comments and Suggestions for Authors
The addressed topic is of particular interest to designers and researchers alike. The manuscript should undergo some changes if it was to be accepted for publication.
Introduction:
This section should be rewritten and expanded. A brief overview of the state of the art should be give. The contribution of the research to solving, at least partly, the identified knowledge gaps should be presented, also.
What do you understand by "strain penetration"?
Figure 1 - one would expect to actually see photos from the failed walls during Chile earthquake. The author should specify that Figure 1 is a schematic representation. Please make sure you have permission to reprint the figure, even though citation was provided.
Section 2.1
Please be consistent when starting a new sentence with "Figure xx". For example, "Figure 1" and "Fig. 2". The manuscript template provides clear guidelines on this matter (similar comment for Figure 3 and Fig. 4).
The author refers to "north and south" directions. Please indicate these directions in Figure 2.
Figure 2 - please make the text visible. Indicate all the components mentioned in the above paragraph .
Figure 3 (caption) - there is mention of "wall 1" and "wall 4". However, the numbering of specimens was not discussed yet and the reader can not understand what they mean.
Page 3, bottom - there is no need to describe the mounting of measurement equipment in so much detail. It would be more useful to actually explain why those locations were chosen as measuring points.
Figure 4 - the text in the figure is difficult to read.
Section 2.3 - this could be moved to become Section 2.1. In this way, a lot of misunderstandings would be avoided related to what actually "wall x" means.
Page 5 - "the idea was to focus...effects" - it would be useful to actually specify the unwanted effects.
Page 6 - it is unclear why the author named the specimens "wall 1".. "wall 4" and then mention "wall 3" is not presented. Please renumber the wall specimens in sequence. The reporting should not follow the same designation of the specimens used in the experimental program. This numbering creates confusion, considering it is not entirely clear why "wall 3" was omitted.
Page 7 - what are the 4 stages the author mentioned and why only stages 1 and 3 were selected?
Page 7 - dimensions of cubes and cylinders are usually given in mm, not m.
Page 8 - do not cite figures from later sections of the manuscript. The figures should be cited in the order they appear in the text. Instead, point to subsequent section.
Page 8 - last sentence of paragraph 1 is not clear. Please use shorter sentences and give full explanation of the considered experimental set-up.
Section 3.1 - the horizontal lines in the loading protocol do not match the description in the text.
Figures 11, 17, 23 - it would be useful to use a, b, c, d, e instead of top left corner, etc. Figure 11e (bottom left) seems similar to 11a (top left). Please double check.
Figures 12, 18, 24 - please mention which part of the figure refers to "south" and which one to "north"
Page 28 - "Fig. 26 shows left" - please change to "Fig. 26left shows.." or use 26a nad 26b instead.
Page 29 - please use an equation editor to write formulas in order to avoid typing errors such as the one related to phi angle.
Conclusions:
This section is too long. Please consider rephrasing it and highlight the main findings of the study.
Comments on the Quality of English LanguageAbstract - "measured were...and spacing" - try reconsider this statement.
Introduction - "however, walls in ... wall section" - the sentence is not clear and contains redundant information. Please rephrase.
Section 2.1 - "a challenge was ...bottom axes" - the statement is not so clear and too long. Please consider shorter sentences.
It is advisable to use short sentences than longer, too complex ones. It would make the conveyed message much clearer to the reader.
Author Response
Dear Reviewer
Thank you for the helpful review. I have several things changed now enhanced in the manuscript. Please find attached my replies.
Kind regards, Harald Schuler

Round 2
Reviewer 1 Report
Comments and Suggestions for Authors
This manuscript has gone a careful revision, and it can be accepted for publication.
Comments on the Quality of English LanguageThis manuscript has gone a careful revision, and it can be accepted for publication.
Author Response
Thank you again for reviewing.
Kind regards
Reviewer 2 Report
Comments and Suggestions for Authors
He reviews the work for the second time. The author introduced a number of additions and explanations. Below I refer to my detailed comments from the first review.
1. Abstract. I propose adding research results in a synthetic way, not only general results.
The abstract was not modified to include detailed conclusions, but I can agree with this approach.
1. Chapter 1. The introduction is very modest. It is proposed to expand the analysis by providing the results of analyzes by other authors, and, above all, to emphasize the purpose of the research and the contribution to the disciplines of Civil Engineering.
Additions have been introduced. I have no comments.
2. Chapter 2.1. In addition to the description of the test stand, a static diagram of the tested walls with the location of the supports used and the location of the loads should be shown. The issue of the uncertainty of vertical loads acting on the wall also requires clarification.
To add to this, I have no comments.
3. Chapter 2.2. Basically, there is no measurement of the shear strain angles, which would help find the correlation between shear force and shear strain. It is also necessary to provide how the curvature was calculated in individual sections.
Clarifications and additions added. I have no comments.
4. Chapter 3. At this point, I suggest adding an addition on how and why the values ​​of horizontal wall loads were different.
I have no comments.
5. Chapters 3.1-3.4. In each model, I propose showing where the original cracks occurred and in which load cycle. I suggest also showing the envelope of the load-displacement relationship.
Added I don't have an explanation. Comments
6. Chapter 4. In addition to flexural deformations, there are also dimensional deformations in the wall, why this aspect was omitted. Of course, the flexural analysis is only valid for flexural deformations caused by the bending moment. There is no information about the dependencies according to which the displacements were calculated taking into account changes in the wall curvature (how the wall curvature was integrated).
Well, maybe the explanations are not very effusive, but the method of analysis adopted is acceptable.
7. In Chapter 5, I propose to add directions for further work.
Added completions. I have no comments.
Author Response
Thank you again for reviewing.
Kind regards.

Reviewer 3 Report
Comments and Suggestions for Authors
I would like to thank the author for taking into account the suggestions made during the reviewing process. Although some of the required changes may have seemed minor, they do play a significant role in better understanding the research and its results.
Figs. 18, 19 & 20 - Figure numbering is kept from the previous version of the manuscript
Page 16, green text - "already explained for..."
In my opinion, the manuscript fulfills all conditions to be published.
Congratulations on your work!
Author Response

(The authors gave the same response as above.)
